# Human miRNAs in Cancer: Statistical Trends and Cross Kingdom Approach

**DOI:** 10.3390/ijms262311594

**Published:** 2025-11-29

**Authors:** Maksym Zoziuk, Vittorio Colizzi, Maurizio Mattei, Pavlo Krysenko, Roberta Bernandini, Fabio Massimo Zanzotto, Stefano Marini, Dmitri Koroliouk

**Affiliations:** 1Interdepartmental Center for Comparative Medicine, Alternative Techniques, and Aquaculture, University of Rome Tor Vergata, Via Montpellier 1, 00133 Rome, Italy; mattei@uniroma2.it (M.M.); dimitri.koroliouk@ukr.net (D.K.); 2Faculty of Medicine, University of Rome Tor Vergata, Via della Ricerca Scientica 1, 00173 Rome, Italy; colizzi@uniroma2.it; 3Department of Biology, University of Rome Tor Vergata, Via della Ricerca Scientica 1, 00173 Rome, Italy; 4Institute of Telecommunications and Global Information Space, National Academy of Sciences of Ukraine, Chokolivskiy bulv. 13, 03186 Kyiv, Ukraine; p.krysenko@gmail.com; 5Department of Clinical Sciences and Translational Medicine, Faculty of Medicine, University of Rome Tor Vergata, Via Montpellier 1, 00133 Roma, Italy; roberta.bernardini@uniroma2.it (R.B.); stefano.marini@uniroma2.it (S.M.); 6Department of Business Engineering “Mario Lucertini”, University of Rome Tor Vergata, Via del Politecnico, 1, 00133 Rome, Italy; fabio.massimo.zanzotto@uniroma2.it; 7Department of Microelectronics, Faculty of Electronics, National Technical University of Ukraine “Igor Sikorsky Kyiv Polytechnic Institute”, Beresteiska Ave. 37, 03056 Kyiv, Ukraine; 8Institute of Mathematics of the National Academy of Sciences of Ukraine, 3, Tereschenkivska St., 01004 Kyiv, Ukraine

**Keywords:** microRNA (miRNA), cross-kingdom comparison, sequence similarity, cancer bioinformatics, plant miRNA

## Abstract

MicroRNAs (miRNAs) are small non-coding RNAs that regulate gene expression post-transcriptionally and are frequently dysregulated in cancer. While most studies focus on individual miRNAs, global patterns and their potential cross-kingdom similarities remain underexplored. This study aims to identify statistically stable human miRNAs in cancer, their key target genes, and analyze sequence complementarity with plant miRNAs to highlight patterns for future research. Experimentally validated human miRNA–gene interactions from miRTarBase were integrated with TCGA expression data across multiple cancers. Using a nonlinear threshold (critical threshold III), 115 underexpressed and 93 overexpressed miRNAs were identified as regulators of 200 genes with the strongest dysregulation. Further, 10,898 plant miRNAs from 127 species were computationally compared to these human miRNAs, and average complementarity scores were calculated to identify plant miRNAs most similar to under- or overexpressed human miRNAs. Statistical parameters such as membership ratios and experiment counts quantified miRNA expression stability. Subsets of human miRNAs exhibited consistent over- or underexpression across cancers, with concordant target gene expression patterns. Several plant miRNAs showed higher complementarity to underexpressed human miRNAs, suggesting reproducible cross-kingdom sequence similarity patterns. Differences in complementarity were modest but systematic, providing a computational framework for prioritizing candidate miRNAs for further study. This work establishes a computational approach integrating human miRNA–gene interactions, cancer expression data, and plant miRNA sequences. It identifies statistically stable miRNAs, key target genes, and cross-kingdom sequence similarities without implying functional or therapeutic activity. The framework can guide future experimental studies in miRNA regulation, comparative genomics, and molecular evolution.

## 1. Introduction

MicroRNAs (miRNAs) are small non-coding RNAs that indirectly regulate DNA replication and protein synthesis. They are found in both plants and animals, are stable, and can be transported to specific locations where they perform their functions [1]. It is believed that plant miRNAs could be introduced into animal or human bodies to compensate for missing or deficient human miRNAs. For effective substitution, plant miRNAs need to have at least 50% sequence similarity with the human miRNAs they aim to activate [2], and replacing one plant miRNA with another requires an even higher similarity of 80–90% [3,4,5]. This biological “complementarity” allows many animal and human miRNAs to be potentially substituted.

Most cancer studies focus on specific individual miRNAs or a limited set of them, analyzing their expression in various tissues or blood samples from healthy and diseased subjects [6,7,8]. It is well known that cancer significantly affects miRNA expression, leading to the hypothesis that regulating miRNA levels could influence cancer progression [9,10]. However, these studies usually address only individual miRNAs in specific diseases.

To better understand overall trends [11] in miRNA expression changes, we used a large database containing miRNA expression data under various conditions and cancer types [12]. Attempts to predict miRNA expression using artificial intelligence, particularly neural networks [13,14,15,16], resulted in accuracy levels not exceeding 70–75%, regardless of sample type or cancer. This suggests that either the mechanisms controlling expression are complex and not fully understood, or multiple factors influence miRNA levels, making accurate prediction difficult.

On the other hand, statistical analysis of miRNA expression distributions across many experiments revealed useful patterns. Each miRNA can be characterized by two parameters: the number of experiments conducted and a “membership ratio” that indicates its tendency toward overexpression or underexpression in cancer. This approach helps identify “special” miRNAs for more detailed analysis.

For applying plant miRNAs therapeutically, it is important to assess the complementarity between human and plant miRNAs. The goal is to find plant miRNAs that closely match human miRNAs that need correction while avoiding those that are already overexpressed. We calculated average complementarity values for different groups and selected plant miRNAs that are best suited to replace deficient human miRNAs.

We also analyzed human miRNA target genes—those regulated by many miRNAs and potentially involved in cancer development. By comparing gene expression data across multiple cancer types [17], we identified key genes and miRNAs important for disease regulation.

This work focuses exclusively on computational and statistical analysis of miRNA sequence complementarity and cancer-associated expression patterns. We aim to identify analytical relationships between human and plant miRNAs, without implying biological transfer or therapeutic use.

## 2. Results

Statistical analysis of 3174 unique miRNAs across 155,417 experiments revealed that approximately 70% of miRNAs consistently belong to either overexpression or underexpression groups, regardless of cancer type or sample source. A neural network trained on miRNA sequences, disease types, and experiment types achieved ~50–65% accuracy in predicting expression direction for all miRNAs, increasing to 70–80% for individual miRNAs with a clear tendency toward over- or underexpression [16]. Filtering by cancer or experiment type had little effect on accuracy. The number of overexpressed and underexpressed miRNAs was nearly equal (1606 vs. 1568), although many miRNAs were represented by only a small number of experiments, and some showed very high membership ratios (>95%) despite limited data. High-throughput expression profiles matched low-throughput literature results in approximately 90% (±5%) of cases.

The results are supported and detailed by multiple data presentations. Table A1 (Appendix B) provides comprehensive expression statistics, the literature references, and established oncological roles for miRNAs, confirming consistency between experimental data and previous research. Table A2 (Appendix B) shows cumulative expression data ordered by growth, highlighting the association of many miRNAs with over- or underexpression in cancer.

Figure 1 illustrates a nonlinear separation of cancer-associated miRNAs. The x-axis shows the membership ratio (Equation (1)), reflecting the difference between the number of studies where a miRNA is upregulated versus downregulated. The y-axis represents the total number of studies. miRNAs located farther from both axes show a stronger association with cancer. Three nonlinear curves indicate a conditional division of miRNAs by their potential relevance (e.g., for prioritization in further research). The color gradient reflects the balance between upregulated and downregulated cases and helps estimate the number of unique miRNAs in each region.

Using nonlinear separation methods, “critical” miRNAs strongly associated with cancer were identified, with the strictest threshold revealing that only a small fraction of these miRNAs have been extensively studied, indicating better coverage for high-concentration miRNAs [18,19,20,21,22,23,24,25,26,27,28,29,30,31,32,33,34,35,36,37,38,39,40,41,42,43,44,45,46,47,48,49,50,51,52,53,54,55,56,57,58,59,60,61,62,63,64,65,66,67,68,69,70,71,72,73,74,75,76,77,78,79,80,81,82,83,84,85,86,87,88,89,90,91,92,93,94,95,96,97,98,99,100,101]. Some miRNAs in Figure 1b correspond to those described in detail in Table A1.

Across all thresholds, underexpressed miRNAs outnumbered overexpressed ones in both diversity and total experiment count, suggesting that in cancer, the number of distinct miRNAs with reduced expression far exceeds those with elevated levels, possibly due to suppression of miRNA production mechanisms, protective body responses, or both. As the number of experiments increases, average membership ratios tend toward zero; however, some miRNAs maintain very high ratios (>0.7) even with 20–50 experiments, confirming the stability of certain expression trends.

Figure 2 presents boxplots of key miRNAs with high membership ratios, demonstrating consistent expression trends. The x-axis shows the logarithmic expression levels across experiments. These miRNAs were selected as the most cancer-relevant, and their average expression across all datasets is plotted to illustrate variability between experiments. This allows evaluation of how strongly expression values fluctuate from one experiment to another.

Each miRNA can regulate a large number of target genes, and identifying specific interaction mechanisms—especially correlations between quantitative characteristics—is an important task. To investigate these relationships, we used the experimentally validated miRNA–gene interactions from the miRTarBase database [102]. For each gene, we calculated the number of targeting underexpressed and overexpressed miRNAs (according to the third nonlinear critical threshold), their difference, total associated miRNAs, and the membership ratio (div).

For deeper analysis, we selected genes targeted by more than 20 miRNAs, reducing the dataset from 7371 to 3495 genes. Figure 3 applies nonlinear separation at the third threshold to distinguish predominantly “downregulated” from “upregulated” genes. The plot follows the same logic as for miRNAs, displaying membership ratio versus the number of miRNAs targeting each gene. This allows assessment of how strongly specific genes depend on miRNA regulation and to what extent this regulation varies. Genes known to be cancer-related (from database [12]) are highlighted, enabling evaluation of whether miRNA influence aligns with genes already established as critical in cancer development.

To assess miRNA dysregulation influence on gene expression in cancer, we integrated these findings with gene expression data from [17], calculating for each gene the mean log fold change (mean_logFC) across cancer types and combining it with miRNA membership ratios. Figure 4 shows a scatterplot dividing genes into quadrants based on the relationship between miRNA expression and target gene expression. Given the regulatory mechanism of miRNAs, high miRNA expression typically corresponds to low target gene expression, and vice versa. Therefore, the second and fourth quadrants—where miRNA and gene expression are inversely correlated—are of primary interest. Genes in these regions follow the expected regulatory pattern and are likely to be predominantly controlled by miRNA rather than other mechanisms. Focusing on these genes provides an effective strategy for uncovering miRNA-driven regulatory processes in oncogenesis.

Focusing on the strongest miRNA-gene regulatory links, we selected 200 genes with the highest combined magnitude of mean_logFC and div. Targeting miRNAs from the third threshold analysis were identified for these genes, yielding 634 underexpressed and 462 overexpressed miRNAs—mostly with a single critical gene target. Restricting to miRNAs targeting at least 10 of these genes resulted in 115 underexpressed and 93 overexpressed miRNAs, which are likely key players in cancer progression due to their pronounced directional expression and regulation of critical genes.

Particularly interesting are genes with positive membership ratios regulated by underexpressed miRNAs, as these may be more amenable to therapeutic restoration compared to suppressing overexpressed miRNAs. Among the 115 underexpressed miRNAs, only 12 overlaps with those previously studied, whereas 35 of 93 overexpressed miRNAs are well characterized, indicating greater research focus on overexpressed miRNAs.

To explore potential interactions between plant miRNAs and critical human miRNAs involved in cancer-related processes, we performed a complementarity analysis between plant miRNAs and the set of critical human miRNAs identified previously. This analysis focused on 93 overexpressed and 115 underexpressed human miRNAs determined using the nonlinear separation line III. Plant miRNA sequences were obtained from the plant miRNA database [104], which contains 10,898 entries from 127 plant species. For each plant miRNA, we calculated the average complementarity score as the sum of complementarity values with all critical human miRNAs divided by the total number of pairwise comparisons. This was done separately for overexpressed and underexpressed human miRNAs.

Table 1 presents the top ten plant miRNAs with the largest absolute differences in mean complementarity scores between 115 and 93 critical human miRNAs. Complementarity coefficients were calculated using Equation (2), and the final column shows their differences. These values indicate which plant miRNAs are statistically more likely to function as potential substitutes for downregulated or upregulated human miRNAs, making them candidates for plant miRNA panels.

Across all plant species, the average complementarity values with underexpressed and overexpressed human miRNAs were similar, differing by less than 0.07 on average. When restricted to human miRNAs associated with the 200 most dysregulated cancer-related genes, certain plant miRNAs exhibited more pronounced differences in complementarity patterns.

These results suggest that sequence-level similarities between certain plant and human miRNAs exist, though the biological or functional significance of these similarities remains to be experimentally established. The analysis presented here is limited to computational comparisons and should not be interpreted as evidence of cross-kingdom regulatory interactions or therapeutic effects.

## 3. Discussion

Key takeaways. The results of the conducted study demonstrate that despite the limitations of neural network models in predicting microRNA expression levels based on nucleotide sequences, statistical methods enable the identification of systemic patterns and correlational relationships between microRNA groups. Similarly, cross-kingdom analysis demonstrates the possibility of systematic assessment and quantification of sequence complementarity patterns. It is essential to emphasize that this study does not postulate functional regulatory activity of plant microRNAs in the human organism. Instead, it offers a bioinformatic perspective on sequence conservation and variability, which may serve as a foundation for subsequent experimental investigations of molecular evolution and RNA-mediated regulation.

Background. In the context of oncological diseases, disruptions in microRNA expression—hyperexpression of oncogenic microRNAs (oncomiRs) and reduced expression of suppressor microRNAs—cause cellular homeostasis imbalance, manifested in alterations of proliferation, apoptosis, migration, and invasion [105]. The vast majority of oncological research focuses on investigating individual microRNAs (particularly miR-21, miR-155) and their specific targets. For instance, miR-21 is characterized by elevated expression in various types of neoplasms and is considered a prototype of oncogenic microRNA [106].

However, global patterns of expression changes—specifically, the existence of a conserved group of microRNAs dysregulated in multiple oncological contexts—as well as comparative analysis between human and non-human microRNAs remain insufficiently studied. A number of review publications emphasize that microRNA dysregulation is not merely a matter of changes in individual molecules but reflects broader transformations of expression profiles in tumor cells. Changes in microRNA biogenesis (e.g., variations in Dicer and Drosha levels) might lead to global alterations in microRNA profiles [107], but at the same time dual functional role of microRNAs as oncogenes and tumor suppressors depends on cellular context, and underscores the significance of their genomic localization in fragile chromosomal regions [108].

Recent years have witnessed growing interest in the phenomenon of cross-kingdom microRNA regulation. Research indicates potential interactions of plant microRNAs with mammalian targets, although this hypothesis remains a subject of scientific debate. A critical review demonstrates that plant microRNAs can be detected in human blood serum and potentially influence human gene expression [109]. Despite reports of potential plant microRNA effects on mammalian organisms, a significant portion of studies is characterized by limited reproducibility of results [110]. In [111] authors present a critical analysis of the evidence base regarding plant microRNA regulation of human genes in the oncological context.

Thus, even with a substantial body of research, there is a deficit of studies dedicated to analyzing the stability of microRNA expression changes across a broad spectrum of oncological nosologies or systematically comparing human microRNAs with microRNAs from other kingdoms by sequence criteria and complementarity potential. This creates space for research integrating large datasets (e.g., microRNA expression from The Cancer Genome Atlas (TCGA)), validated miRNA-target interactions (particularly from miRTarBase), and cross-kingdom comparative analyses.

Conservation of expression changes. It has been established that a significant proportion of microRNAs (both upregulated and downregulated) demonstrate a conserved direction of expression changes regardless of histological tissue type and nosological cancer form. These microRNAs may play a critical role in the mechanisms of resistance and progression of oncological diseases or, conversely, in the organism’s antitumor responses.

miRNAs and target genes. Considering that the directions of expression changes in microRNAs and their target genes do not always correlate, one can hypothesize the dominance of alternative mechanisms of gene expression regulation. However, in cases where expression change directions coincide, it may be hypothesized that these specific microRNA-target gene interactions play a critical regulatory role.

Homology. Despite the fact that the vast majority of critical human microRNAs do not demonstrate significant homology with plant microRNAs, isolated cases of high degrees of similarity can be identified. Such plant microRNAs may possess substantial regulatory potential provided that their human analogs exert pronounced effects on key oncogenes or tumor suppressor genes, alterations in the expression of which can initiate cascading effects on oncogenesis.

Importance. Complete accounting of all microRNA-mediated regulatory mechanisms represents an exceptionally complex task. However, a statistical approach enables assessment of cumulative effects on multiple regulatory mechanisms through modulation of the expression of human microRNAs characterized by consistently high or differential expression in the human organism. Having identified such critical microRNAs, it becomes possible to establish complexes of plant microRNAs based exclusively on sequence homology.

A promising direction involves identification of specific plant species under particular environmental conditions and locations (as microRNA concentrations in a single species may vary depending on environmental conditions [112]) containing the necessary set of microRNAs. By investigating the bioavailability of these microRNAs to the human organism, the potential of a given plant to modulate oncological disease development can be established. It is important to note that individual critical oncogenes or tumor suppressor genes may be targets of entire sets of regulatory microRNAs, opening an alternative approach to establishing plant microRNA complexes for targeting specific regulatory genes. The research process encompasses a range from establishing critical human and plant miRNAs to identifying key genes with high sensitivity to microRNA-mediated regulation (as opposed to other regulatory mechanisms).

Limitations and Perspectives. The primary impediment relates to the restricted number of plant miRNAs (documented in databases) demonstrating high homology with a significant portion of critical human microRNAs in oncological diseases. The effect of an individual microRNA on a specific gene may encounter numerous barriers to achieving a detectable effect; therefore, the fundamental strategy consists of a comprehensive approach—selection of a sufficient number of microRNAs that will synergistically target multiple highly regulated genes. It is anticipated that significant therapeutic effects may be achieved up to the development of broad-spectrum drugs for oncological diseases.

Priority directions for future research include conducting experiments to assess the effects of critical human microRNAs in vitro and in vivo, with monitoring of key parameters: cell viability, changes in gene and microRNA expression profiles, as well as general characteristics of tumor process progression. A critically important direction involves identification of plant microRNAs with high homology to critical human microRNAs (preferably experimentally validated) and establishment of plant species containing complexes of such microRNAs. It is essential to ensure cost-effectiveness and rapidity of this process, considering the large number of potential candidates and variability of their characteristics across different parameters. The final stage consists of bioavailability assessment and development of strategies to enhance the accessibility of these microRNAs to the human organism.

## 4. Materials and Methods

The study was based on a publicly available miRNA expression database containing 3174 distinct miRNAs across 155,417 experiments, with records including cancer type, experiment type, logFC values, and additional parameters such as T/B values and mean expression levels [12]. Data originated from multiple sources and covered various cancer types and sample origins (tissue, blood, cells).

The research consisted of two main stages. First, a feed-forward neural network was trained to predict logFC values and expression direction (“up” or “down”) using mature and pre-miRNA sequences, disease type, experiment type, and other metadata. Numerical experiments were conducted both with and without filtering by cancer type or experiment type [16].

Second, a statistical aggregation was performed for each miRNA, calculating the cumulative logFC across all experiments, the number of underexpression and overexpression cases, and the “membership ratio” defined as: (1)$$div=\frac{{num}_{{of}_{down}}-{num}_{{of}_{up}}}{{num}_{{of}_{down}}+{num}_{{of}_{up}}}$$


This allowed classification of miRNAs into over- or underexpressed groups and the identification of “critical” miRNAs using nonlinear separation curves. Filtering options enabled focused analysis for specific cancers or experiment types, and results from high-throughput data were validated against low-throughput literature data, achieving ~90% (±5%) agreement depending on filters.

For the analysis, data on human miRNAs [103] and their experimentally confirmed target genes were obtained from the miRTarBase database (release 9.0) [104], which contains curated information from experimental studies. Only verified miRNA–gene interactions were included; predicted interactions were excluded to ensure reliability. Differentially expressed miRNAs were determined using a predefined nonlinear “critical threshold III,” which separates underexpressed and overexpressed miRNAs in cancer samples.

For each gene, the total number of targeting underexpressed and overexpressed miRNAs was calculated, along with their difference, the total number of miRNAs, and the membership ratio (*div* parameter), reflecting the relative predominance of under- or overexpressed regulators. Genes with fewer than 20 targeting miRNAs were excluded from further analysis, reducing the dataset from 7371 to 3495 genes.

Mean gene expression values (logFC) across multiple cancer types were obtained from The OncoDB database [17]. For each gene, the mean logFC across all cancer types was calculated, providing a single expression value per gene. These values were combined with the miRNA-derived membership ratio to produce two-dimensional distributions for subsequent interpretation.

Critical genes were defined as those with the highest combined absolute values of mean logFC and membership ratio, resulting in a set of 200 genes. The miRNAs targeting these genes were then re-examined, and only those regulating at least 10 critical genes were selected. This yielded 115 underexpressed and 93 overexpressed miRNAs for detailed analysis.

Sequences of plant miRNAs were obtained from the plant miRNA database [105], which contains 10,898 entries representing 127 plant species. Human miRNAs were taken from the dataset described in the previous section, where critical miRNAs were identified using the nonlinear separation line III. Two human miRNA sets were used: The full set of 786 underexpressed and 607 overexpressed miRNAs from the III threshold analysis.A selective set of 115 underexpressed and 93 overexpressed miRNAs that target the 200 most critical cancer-related genes.

Complementarity between each plant miRNA and each human miRNA was calculated using the alignment and similarity scoring method described in [113]. The average complementarity score for each plant miRNA was computed as:(2)$$AvgComp=\frac{\sum_{i=0}^{N}Comp(p,\;h_{i})}{N}$$where *p* is the plant miRNA, $h_{i}$ is the *i*-th human miRNA from the selected set, and *N* is the number of comparisons. This score was calculated separately for underexpressed and overexpressed human miRNAs.

For each plant miRNA, the difference in average complementarity between underexpressed and overexpressed human miRNAs was determined. Plant miRNAs with the largest absolute differences were considered potential candidates for preferential targeting of underexpressed miRNAs. Separate rankings were produced for the full set of human miRNAs and for the selective set associated with critical genes.

All sequence handling, complementarity calculations, statistical analyses and visualizations (including threshold-based plots, membership ratio graphs, and boxplots) were performed using Python version 3.11, with custom scripts developed in PyCharm IDE 2022.1 for pairwise alignment and score aggregation.

## 5. Conclusions

This study presents a computational framework integrating validated human miRNA–gene interactions with gene expression data and comparative sequence analysis involving plant miRNAs. The results identify statistically significant trends in sequence complementarity patterns. This study reveals that around 70% of miRNAs show consistent over- or underexpression across cancer types, suggesting stable regulatory roles. While predictive models using sequence and metadata achieved moderate accuracy, strong trends in individual miRNAs improved prediction significantly. Underexpressed miRNAs were more diverse and more frequently linked to cancer-related gene regulation than overexpressed ones. A small subset of critical miRNAs was identified, many of which remain understudied, especially among underexpressed types. Expression data matched literature findings in most cases, confirming reliability. Integration with gene expression profiles highlighted key miRNA–gene relationships, with potential therapeutic value, particularly in restoring underexpressed miRNAs. Complementarity analysis with plant miRNAs showed modest sequence-level similarity to human miRNAs, warranting further investigation.

The approach highlights how large-scale bioinformatics can help detect conserved miRNA features and prioritize candidates for future laboratory validation.

## Figures and Tables

**Figure 1 ijms-26-11594-f001:**
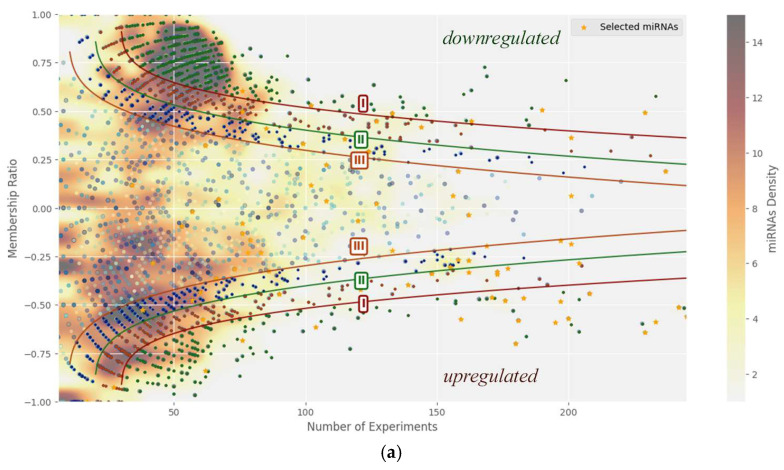
Nonlinear separation of critical miRNA. (**a**) Distribution plot with three nonlinear curves and concentration spectrum of individual miRNAs. (**b**) Distribution graph with a nonlinear curve (I) and miRNA from the literature. All the dividing lines were taken based on our considerations only. The first line is the main one and highlights critical miRNAs that are important from the point of view of studying their dependence on the presence of cancer.

**Figure 2 ijms-26-11594-f002:**
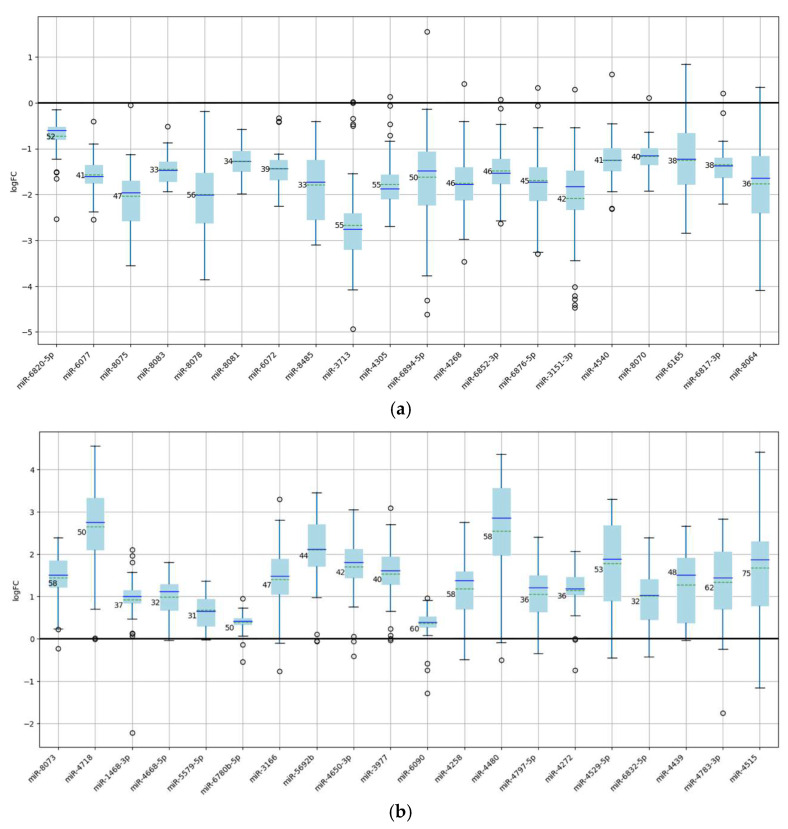
Boxplots for the miRNA with the highest membership ratios. (**a**) 20 underexpressed miRNAs. (**b**) 20 overexpressed miRNAs.

**Figure 3 ijms-26-11594-f003:**
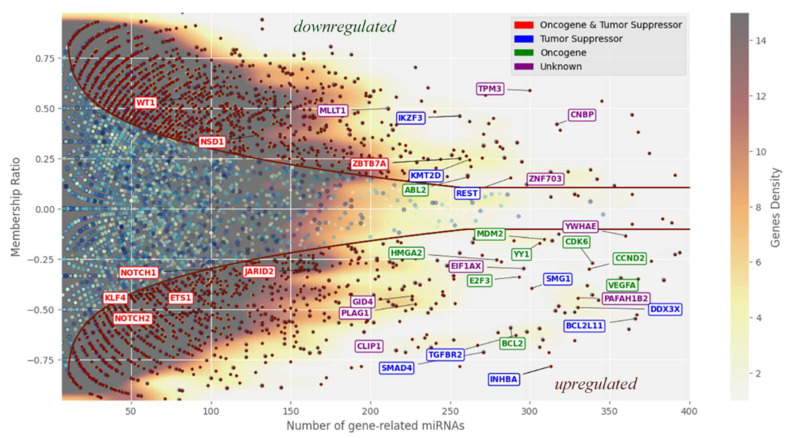
Graph of membership ratio dependence on the number of miRNAs for genes that are targets for underexpressed and overexpressed miRNA (number of which is less than 20 for each gene). Only a small part of the genes for which one or another effect on the development of cancer has been proven is highlighted here (in text).

**Figure 4 ijms-26-11594-f004:**
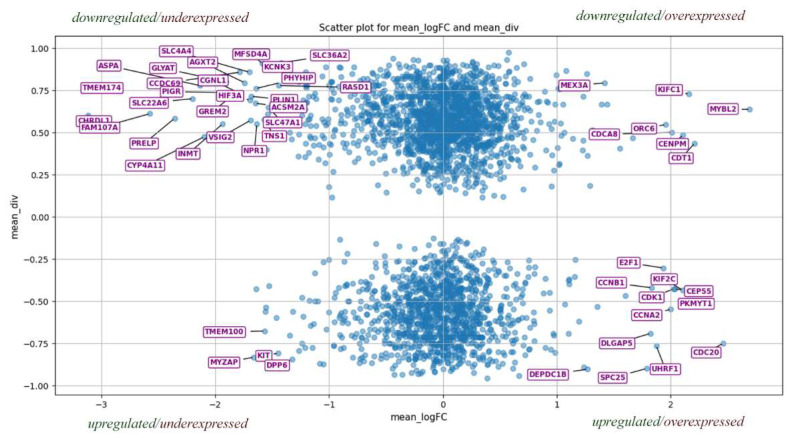
Location of genes according to mean logarithmic expression (mean_logFC) for all types of cancer and membership ratio (mean_div) based on miRNA expressions. The 50 most distant genes from the center are shown here as an example, although the number of interesting genes is not limited to this number. Each quarter represents a separate piece of information. The first is up-regulated miRNAs (for this gene) and under-expressed gene. The second—downregulated miRNAs, but overexpressed gene. The third is upregulated miRNAs, but an underexpressed gene. The fourth is upregulated miRNAs and underexpressed genes. Those genes that are closer to zero by mean_logFC have zero expression. This plot was first presented in [103].

**Table 1 ijms-26-11594-t001:** Plant miRNAs which are candidates for plant miRNA panels. The first column is the name of miRNA. The second (and third) column is the similarity coefficient – Equation (2) for over and underexpressed miRNAs. The third column is the difference between them.

miRNA	Coefficient_Similarity_Up	Coefficient_Similarity_Down	Diff
ptc-miRf12120-akr	0.567818	0.680251	0.112433
cre-miR914	0.460989	0.573858	0.11287
ptc-miRf10479-akr	0.493849	0.607797	0.113949
ptc-miRf10488-akr	0.529684	0.645046	0.115362
ptc-miRf10495-akr	0.486887	0.603567	0.11668
osa-miR531b	0.459638	0.576326	0.116688
tae-miR2019	0.467059	0.584464	0.117405
peu-miR2911	0.412766	0.542227	0.129461
osa-miR1848	0.442845	0.57365	0.130805
osa-miR531	0.495555	0.635791	0.140237

## Data Availability

All the newly (and additional) generated data are presented in Appendix A.

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
