# Peer review of "Human miRNAs in Cancer: Statistical Trends and Cross Kingdom Approach"

_ijms, 2025, doi:10.3390/ijms262311594_

Round 1
Reviewer 1 Report
Comments and Suggestions for Authors
The article titled “Bioinformatic comparison of human and plant miRNAs in cancer” presents a thorough computational analysis of human and plant miRNAs in the context of cancer. It provides a statistically grounded framework for cross-kingdom sequence analysis and miRNA–gene regulatory characterization. By integrating large miRNA expression datasets from TCGA and miRTarBase, the study identifies cancer-relevant miRNAs and their target genes using rigorous nonlinear thresholding and membership ratio assessment.
The authors validate their findings against literature sources, showing about 90% agreement, and clearly distinguish between over- and underexpressed miRNAs in cancer, highlighting the predominance and diversity of underexpressed miRNAs. Their computational pipeline for pairing plant and human miRNAs is comprehensive, using average complementarity scores across thousands of plant miRNAs to reveal subtle, reproducible trends in cross-kingdom similarity.
Notably, the miR-34 family is a key tumor-suppressor miRNA group in humans, regulating cell cycle, apoptosis, and metastasis as highlighted in https://www.cell.com/molecular-therapy-family/nucleic-acids/fulltext/S2162-2531(24)00080-5. Some plant-derived miRNAs, such as miR-159 and miR-156, have been proposed to target human oncogenes or tumor pathways. Comparing these plant miRNAs with miR-34 can reveal functional or sequence similarities relevant to cancer modulation, highlighting the importance of human tumor-suppressive miRNAs in bioinformatic analyses involving cross-kingdom comparisons. The authors are encouraged to include this aspect.
Figures and tables are well-organized, effectively illustrating classification boundaries, data distributions, and candidate miRNAs. The manuscript avoids overstating functional or therapeutic relevance, emphasizing that the results are purely computational. The discussion acknowledges the limitations of neural network predictions, stresses the value of multi-experiment statistical aggregation, and points to future experimental validation.
Overall, the study offers a clear methodology for miRNA prioritization and comparative genomics. Minor issues remain, such as clarity in some sections, deeper data presentation, and additional discussion on functional impacts and specific gene–miRNA interactions. The Materials and Methods are detailed enough for reproducibility but lack benchmarking of comparison algorithms and explanation of plant miRNA selection. The analysis would also benefit from more discussion on potential mechanisms of cross-kingdom regulation and the clinical relevance of highlighted miRNAs.
In summary, this article provides a large-scale computational resource for miRNA research and generates new hypotheses for experimental exploration. With modest improvements in clarity, context, and presentation, the manuscript could achieve maximum impact.
Author Response
Comments 1: Notably, the miR-34 family is a key tumor-suppressor miRNA group in humans, regulating cell cycle, apoptosis, and metastasis as highlighted in https://www.cell.com/molecular-therapy-family/nucleic-acids/fulltext/S2162-2531(24)00080-5. Some plant-derived miRNAs, such as miR-159 and miR-156, have been proposed to target human oncogenes or tumor pathways. Comparing these plant miRNAs with miR-34 can reveal functional or sequence similarities relevant to cancer modulation, highlighting the importance of human tumor-suppressive miRNAs in bioinformatic analyses involving cross-kingdom comparisons. The authors are encouraged to include this aspect.
Response 1: In our task the main aim was to make conceptual investigation across all the human and plant miRNA by calculating mean similarity values. Focusing on specific interaction between human and plant miRNAs wasn`t our main task, but we absolutely agree that this is should be considered as huge additional information to our article, but gather and put all this information and make sure nothing lost can take massive amount of space. Thank you for an notion, we are working on specific analysis of miRNAs that are taken from out primary investigation that is presented in this manuscript.
Reviewer 2 Report
Comments and Suggestions for Authors
In the present work, Zoziuk et al. compare miRNA expression in human cancer and 127 plants. Despite the comparison and the background can be interesting i feel that it cannot being considered for publication in its current form. A deep (or at least a minimum) discussion of the findings, implications, limitations and future directions should be provided. Also a greater development of the discussion should also be required in order to understand the rationale of the work.
• What is the main question addressed by the research?
How are the sequence complementarity between miRNAs from human tumoral cells and those derived from plants
• Do you consider the topic original or relevant to the field? Does it address a specific gap in the field? Please also explain why this is/ is not the case.
Yes, it is original and addresses a specific gap relating the meaning and evolution of miRNAs in cell biology.
• What does it add to the subject area compared with other published material?
A broader exploration of the complementarity between tumor and plant miRNAs
• What specific improvements should the authors consider regarding the methodology?
For the objectives are fine
• Are the conclusions consistent with the evidence and arguments presented and do they address the main question posed? Please also explain why this is/is not the case.
In this point i feel that there are some important points to be addressed, specially regarding the meaning, limitations, significance, background, potential applications and review previous works conducted in this field.
• Are the references appropriate?
Yes
• Any additional comments on the tables and figures.
No additional comments
Author Response
Comments 1.
A deep (or at least a minimum) discussion of the findings, implications, limitations and future directions should be provided. Also a greater development of the discussion should also be required in order to understand the rationale of the work.
Response 1. Thanks for the comment. We have greatly improved the discussion section, adding the necessary subsections and links.
Reviewer 3 Report
Comments and Suggestions for Authors
The manuscript by Zoziuk et al. presents a comprehensive bioinformatic analysis exploring the relationship between human miRNA dysregulation in cancer and sequence complementarity with plant miRNAs. The study is ambitious in scope, integrating large-scale miRNA expression data, gene interaction networks, and cross-kingdom sequence comparisons. The authors provide a robust computational framework and clearly state the limitations of their approach, emphasizing that their findings are purely bioinformatic and do not imply functional cross-kingdom activity.
However, a major weakness of this study is its limited practical or translational value. The dietary and/or therapeutic potential of the specific plant species harboring the identified microRNAs is neither covered in the experimental design nor discussed in the manuscript. The discussion section has significant potential for expansion, particularly by incorporating existing literature on plant and other exogenous microRNAs effects on cancer.
Other issues:
- The current title, "Bioinformatic Comparison of Human and Plant miRNAs in Cancer," is somewhat confusing, as it may imply a functional role for plant miRNAs in human cancer. A more precise title would improve clarity.
- Figures 1–4 are referenced but not fully described in the text. Their captions are minimal, and the figures themselves are not included in the provided text. A more detailed explanation of what each figure represents would improve readability.
- Table 2 is informative but would benefit from a clearer explanation of how the "coefficient_similarity" was calculated and what the values represent (e.g., alignment scores, percent identity).
- The manuscript contains a few typographical errors (e.g., "Ricerca Scientieca" should be "Ricerca Scientifica").
- Some references in the appendix (e.g., Table A1) are incomplete or unclear (e.g., "unknown" oncological effect for well-studied miRNAs like miR-21-5p).
Author Response
Comments 1: However, a major weakness of this study is its limited practical or translational value. The dietary and/or therapeutic potential of the specific plant species harboring the identified microRNAs is neither covered in the experimental design nor discussed in the manuscript. The discussion section has significant potential for expansion, particularly by incorporating existing literature on plant and other exogenous microRNAs effects on cancer.
Response 1: Thanks for a comment. The primary goal of the article was to establish statistical dependencies that would enable the identification of specific human or plant miRNAs or genes that warrant attention when initiating experimental research. Regarding the Discussion section, we completely agree with your feedback. Our initial concern was that the article's total length might become excessive. However, we recognize the value of your suggestion and agree that the Discussion section indeed needs to be expanded. Therefore, we have now revised and expanded the Discussion accordingly to provide a more comprehensive context for our findings.
Comments 2: The current title, "Bioinformatic Comparison of Human and Plant miRNAs in Cancer," is somewhat confusing, as it may imply a functional role for plant miRNAs in human cancer. A more precise title would improve clarity.
Response 2: Thanks for an comment. We suggest that this new name of an article might be appropriate more: Human miRNAs in Cancer: Statistical Trends and Plant miRNA Similarities.
Comments 3: Figures 1–4 are referenced but not fully described in the text. Their captions are minimal, and the figures themselves are not included in the provided text. A more detailed explanation of what each figure represents would improve readability.
Response 3: Thanks for a comment, we have added a more extensive explanation of the pictures and tables in the text.
Comments 4: Table 2 is informative but would benefit from a clearer explanation of how the "coefficient_similarity" was calculated and what the values represent (e.g., alignment scores, percent identity).
Response 4: Thanks for a comment, we added a more extensive explanation of Table 2 in the caption itself.
Comments 5: The manuscript contains a few typographical errors (e.g., "Ricerca Scientieca" should be "Ricerca Scientifica").
Response 5: Thanks for a comment, but unfortunately, we couldn’t find any.
Comments 6: Some references in the appendix (e.g., Table A1) are incomplete or unclear (e.g., "unknown" oncological effect for well-studied miRNAs like miR-21-5p).
Response 6: Thanks for a comment, but mir-21-5p was marked as oncomir in table A1. Also, has-mir-21-5p placed in table A1 twice, so we changed that into correct way. We checked all the other miRNAs for recent studies about their potential effects, and changed their cancer effect «affiliation».
Round 2
Reviewer 3 Report
Comments and Suggestions for Authors
the manuscript has been improved
Author Response
Thank you for a comment. We fixed typo in affiliations ("Scientifica").